# Key Role of Corncob Based-Hydrochar (HC) in the Enhancement of Visible Light Photocatalytic Degradation of 2,4-Dichlorophenoxyacetic Acid Using a Derivative of ZnBi-Layered Double Hydroxides

**DOI:** 10.3390/ma16145027

**Published:** 2023-07-16

**Authors:** Ngo Thi Tuong Vy, Dang Nguyen Nha Khanh, Nguyen Ngoc Nghia, Le Hai Khoa, Pham Tuan Nhi, Le Xuan Hung, Doan Thi Minh Phuong, Nguyen Thi Kim Phuong

**Affiliations:** 1Institute of Applied Materials Science, Vietnam Academy of Science and Technology, Ho Chi Minh 700000, Vietnam; tuongvyngo93@gmail.com (N.T.T.V.); dnnhakhanh@gmail.com (D.N.N.K.); nghia10194@gmail.com (N.N.N.); 2Vietnam Academy of Science and Technology, Graduate University of Science and Technology, Hanoi 100000, Vietnam; lehaikhoa87@gmail.com; 3Institute for Tropical Technology, Vietnam Academy of Science and Technology, Hanoi 100000, Vietnam; 4Hochiminh City Institute of Resources Geography, Tay Nguyen Institute of Scientific Research, Vietnam Academy of Science and Technology, Ho Chi Minh 700000, Vietnam; ptnhi@hcmig.vast.vn; 5Institute of Research and Development, Duy Tan University, Da Nang 550000, Vietnam; lexuanhung@duytan.edu.vn; 6Faculty of Chemical Engineering, Ho Chi Minh City University of Industry and Trade, Ho Chi Minh 100000, Vietnam; phuongdtm@hufi.edu.vn

**Keywords:** HC-ZnBi-LDO, heterojunction, 2,4-dichlorophenoxyacetic acid, visible light exposure, kinetic degradation

## Abstract

A superior heterojunction of HC-ZnBi-LDO was synthesized in two steps, namely hydrothermal carbonization, followed by co-precipitation. The 2% HC-ZnBi-LDO heterojunction photocatalysts could degrade over 90.8% of 30 mg/L 2,4-dichlorophenoxyacetic acid (2,4-D) using 1.0 g/L of the catalyst after 135 min of visible light exposure at pH 4. The activity of 2% HC-ZnO-LDO was remarkably stable. Approximately 86.4–90.8% of 30 mg/L 2,4-D was degraded, and more than 79–86.4% of TOC was mineralized by 2% HC-ZnBi-LDO at pH 4 after 135 min of visible light exposure during four consecutive cycles. The rapid separation and migration of charge carriers at the interfaces between HC and ZnBi-LDO were achieved within 2% HC-ZnBi-LDO. Moreover, the electron acceptor characteristic of HC in 2% HC-ZnBi-LDO caused the recombination of charge carriers to decrease significantly, thus generating more reactive radicals, such as hydroxyl radicals (OH^●^) and superoxide radicals (O_2_^●−^). These results demonstrate that the novel 2% HC-ZnBi-LDO is a superior photocatalyst for the remediation of hazardous organic pollutants.

## 1. Introduction

Hydrochar (HC) is considered a promising engineering solution with applications in water treatment and promotes the circular economy concept. As an inexpensive material, HC is produced from a wide variety of large waste biomasses (e.g., agricultural, forestry, food waste sludge) through low-energy hydrothermal carbonization (HTC) [1]. The advantage of HTC over pyrolysis is the direct treatment of wet waste biomass, especially those with a high moisture content. HTC is performed at relatively lower temperatures, typically from 80 to 240 °C, under subcritical water pressure [2]. Compared with biochar (BC), HC research is still in its infancy in terms of the optimization of production processes and its water treatment potential. Therefore, research on converting different types of waste biomass into valuable HC products requires further development, which should include the optimization of the production parameters to realize its full potential. HC is considered an ideal carrier that improves the removal efficiency of targeted pollutants; it is gaining popularity as an economically and environmentally sustainable material with many applications, including being used as an effective adsorbent for the treatment of contaminated water [2]. HC is a product of the biomass hydrolysis reactions that occur in the HTC process; the surface of HC contains various chemically reactive functional groups, including acidic functional groups, which can promote the adsorption of pollutants through electrostatic interactions [3]. However, HC is produced with a low surface area when HTC is used as the sole manufacturing process; therefore, it may be necessary to combine HTC with additional activation steps to effectively change the physicochemical properties, optimize the specific functional groups, and increase the porosity and surface area of HC to increase the adsorption capacity or selective removal of pollutants [4,5,6,7,8,9].

The outstanding feature of advanced oxidation processes (AOPs) (e.g., photocatalysis, Fenton oxidation, electro-Fenton/ultrasound, etc.) in the removal of organic pollutants is the generation of strong reactive radical species [10,11]. Among the existing treatment technologies, photocatalysis is emerging as a green technology to remove organic waste from contaminated water and involves light-induced reactions that occur in the presence of a catalyst. A highly efficient photocatalyst must have a narrow band gap and a suitable surface defect state. The narrow band gap allows the photocatalyst to absorb both UV and visible light, and the surface defect state facilitates the entrapment of the photoinduced electrons (e^−^) or holes (h^+^) to prevent their recombination. The rapid recombination of charge carriers and conduction band electrons and valence band holes prevents reactions with O_2_ and H_2_O, which generate superoxide radicals (O_2_^●−^) and hydroxyl radicals (OH^●^), respectively [12]. OH^●^ and O_2_^●−^ are the main factors in the breakdown of persistent pollutants from wastewater because of their strong oxidizing capacity [12]. 

Semiconductors are widely used as catalysts because of their simplicity, ease of synthesis, and high efficiency. Titanium, bismuth, zinc, and tin are the preferred elements for the synthesis of semiconductors. Recently, bismuth-based semiconductors have shown promise as advanced photocatalyst materials due to their high chemical stability and excellent transparency for light penetration. Many researchers have focused on the design of highly efficient photocatalysts under visible light. One approach is to introduce heteroatoms into the lattice sites of a semiconductor, and another is to fabricate new photocatalysts with adjustable bandgaps from two or more semiconductors. To date, many heterostructures, such as FeNi_3_/SiO_2_/CuS [10], Bi_2_Sn_2_O_7_–reduced graphene oxide (rGO) [13], Bi_2_S_3_–Bi_2_Sn_2_O_7_ [14], TiO_2_@MgFe_2_O_4_ [15], ZnBi_2_O_4_–Bi_2_S_3_ [16], ZnBi_2_O_4_–graphite [16], ZnBi_2_O_4_–C_3_N_4_ [17], Eu-doped Bi_2_WO_6_ [18], g-C_3_N_4_/C-dots [19], rGO–ZnBi_2_O_4_-Bi_2_S_3_ [20], and rGO–ZnBi_2_O_4_ [21], have been constructed and used as a promising candidate for the remediation of toxic organic pollutants.

As a derivative of layered double hydroxides (LDHs) with low recombination rates of charge carriers, mixed-metal oxides are considered sustainable and promising photocatalysts for application in hazardous pollutant remediation. As a topical semiconductor and promising photocatalyst, ZnBi-LDO, one of the simplest mixed-metal oxides derived from LDHs, has been the focus of several research groups [16,17,20] owing to its high stability, small optical band gap, and low conduction band edges, which allow it to make better use of sunlight. 

In this study, a highly efficient HC-ZnBi-LDO heterojunction was synthesized in two steps, namely hydrothermal carbonization followed by co-precipitation. The close association between the ZnBi-LDO semiconductor and HC improved photocatalytic activity. The fabricated HC-ZnBi-LDO heterojunction was used to degrade 2,4-D with the participation of visible light. Further, the influences of various operating parameters, such as the pH of solution, catalyst dosage, and initial 2,4-D concentration, were considered to achieve the optimal decomposition of 2,4-D. The outstanding photocatalytic performance of the HC-ZnBi-LDO material obtained from this study may bring a new prospect to the research on the fabrication of semiconductor photocatalysts to improve photocatalytic activity.

## 2. Materials and Methods

### 2.1. Materials

All chemicals, such as 2,4-D, Bi(NO_3_)_3_·5H_2_O, Zn(NO_3_)_2_·6H_2_O, HNO_3_, NaOH, Na_2_EDTA, *tert*-butanol and *p-*benzoquinone, were of analytical grade without further purification and were purchased from Sigma Aldrich (St. Louis, MO, USA).

### 2.2. Preparation of Photocatalysts (ZnBi-LDO and HC-ZnBi-LDO)

First, approximately 50 g of finely ground corncobs were loaded into the reaction vessel with 200 mL of 2 M KOH and stirred at 200 rpm for 30 min before sealing the reaction vessel. The reactor was then kept at 180 °C for 6 h. The resulting solid material was rapidly cooled down to room temperature before being calcined in a stream of nitrogen at 700 °C for 2 h. Next, the calcined material was washed with distilled water, dried at 105 °C for 24 h, and denoted HC. 

Second, an aqueous solution composed of Zn(NO_3_)_2_·6H_2_O and Bi(NO_3_)_3_·5H_2_O in HNO_3_ (5%) with a molar ratio of 3:1 was added drop by drop to a 1 M NaOH solution containing different amounts of HC. The procedure was carried out in an ultrasonic bath at 100 W irradiation, and the solution was kept at pH 10 throughout the process. The suspension was then incubated in an ultrasonic bath at 100 W of irradiation for 24 h. The obtained solid samples were separated by centrifugation and washed with deionized water, then dried at 70 °C for 10 h, followed by calcination at 350 °C for 3 h to obtain HC-ZnBi-LDO. Hereafter, it will be called ZnBi-LDO, 1% HC-ZnBi-LDO, 2% HC-ZnBi-LDO, and 5% HC-ZnBi-LDO.

### 2.3. Characterization

Crystalline nature, phase composition, and phase identification were conducted using a Rigaku Ultima IV X-ray diffractometer (Tokyo, Japan) using Cu Kα radiation (λ = 1.54051 Å) operated at 40 mA current and 40 kV voltage. The Fourier transform infrared (FT-IR) spectrum was carried out in a KBr dispersion in the range of 4000–400 cm^−1^ using a Nicolet Magna-560 FT-IR spectrometer (Thermo, Waltham, MA, USA). The band gap energy (Eg) of the samples was identified using V670 UV-Vis diffuse reflectance spectroscopy (Jasco, Tokyo, Japan). Field Emission Scanning Electron Microscope (FE-SEM) equipped with energy-dispersive X-ray spectroscopy (EDS) was used to determine the images and elemental composition of the samples (Hitachi S-4800, Tokyo, Japan). A Horiba system (FL3-22-1551C-3012-FL, Kyoto, Japan) was employed to record the photoluminescence (PL) spectra of the photocatalysts. X-ray photoelectron spectroscopy (XPS) was performed using the Nexsa XPS system (ThermoFisher Scientific, Abingdon, UK). TOC-SSM5000A and TOC-VCPH analyzers from Shimadzu (Kyoto, Japan) were used for the determination of organic carbon (TOC) in solid and liquid samples, respectively.

### 2.4. Photocatalytic Experiment

The visible light photocatalytic performance was monitored based on 2,4-D decomposition. The batch reactor was maintained at a constant temperature and stirred frequently for good dispersion of the catalyst. Before turning on visible light, which was provided by a 300 W halogen lamp (Osram, Munich, Germany), the suspension was allowed to stir continuously in a dark room for 60 min to establish an equilibrium between 2,4-D and the catalyst surface. At regular intervals over a total of 135 min of light exposure, 5 mL aliquots were collected, and the change in concentration of 2,4-D in solution was traced by the UV-Vis spectrophotometer (Lamda XLS+, Perkin Elmer, Waltham, MA, USA). 

## 3. Results and Discussion

### 3.1. Material Characterization

*XRD:* Figure 1a shows the XRD patterns of pristine HC, ZnBi-LDO, and HC-ZnBi-LDO, comprising different HC percentage loadings. For the XRD profiles of HC, there were two main peaks located at 23.2 and 44.5°, which match well with the (002) and (100) planes of amorphous carbon, respectively [22]. For pristine ZnBi-LDO, there were a few principal peaks located at the 2θ position of 27.4°, 32.8°, 45.3°, and 47°, which were closely associated with tetragonal zinc bismuth oxide (JCPDS No. 043-0449); moreover, there were characteristic peaks located at the 2θ position of 31.7°, 34.6°, 36.3°, 56.5°, and 62.9°, which were associated with hexagonal zinc oxide (JCPDS No. 079-0207). In addition, there was a characteristic diffraction peak for bismuth hydroxide centered at 30.3° (JCPDS No. 001-0898). For HC-ZnBi-LDO comprising different HC percentages, the new diffraction peak located at the 2θ position of 23.7° indicates the presence of HC in the structure. A diffraction peak located at 23.7° corresponded to the (002) crystallographic diffraction of hexagonal bulk graphite with a lattice spacing between the plane (*d_hkl_*) of 3.72 Å [23,24,25].

*FT-IR:* FT-IR spectroscopy was performed to identify the active functional groups present in ZnBi-LDO and HC-ZnBi-LDO, comprising various percentages of HC. As seen in Figure 1b, in all the spectra, a broad absorption peak at 3410 cm^−1^ correlated with the water molecule (H–O–H), which was adsorbed by the materials. [26]. For ZnBi-LDO and HC-ZnBi-LDO, the peaks at 1385 cm^−1^ and 848 cm^−1^ were indexed for the stretching vibrations of Bi–O and Bi–O–Bi, respectively [27]. For HC, the peaks located at approximately 1560 and 1015 cm^−1^ corresponded to the stretching bend of C=C and the stretching vibration of C-O-C, respectively. The peak located at approximately 2922 cm^−1^ was indexed to the aliphatic C-H vibration. The results indicate that an aromatization process occurs during the HTC process, during which the polysaccharide is converted into HC. However, these bands were not clearly visible in the spectrum of HC-ZnBi-LDO.

*Surface study:* The surface morphology and structure information of the prepared samples are shown in Figure 1c. The SEM image of the HC had a relatively coarse, porous, and fibrous surface, while pristine ZnBi-LDO consisted of unevenly stacked particles. The SEM images of the HC-ZnBi-LDO samples revealed that ZnBi-LDO was uniformly bound to HC. Furthermore, the EDS spectrum of 2% HC-ZnBi-LDO showed that elements including Zn, Bi, O, N, and C were evenly distributed over the material.

### 3.2. Photodegradation Process

#### 3.2.1. Effect of Hydrochar Source Loadings on Heterojunction HC-ZnBi-LDO

The effect of the HC source in HC-ZnBi-LDO on the degradation of 2,4-D was carried out by dispersing 50 mg of each catalyst in 50 mL of solution containing 30 mg/L 2,4-D at a pH ≈ 4.0. As shown in Figure 2a, the decomposition of 2,4-D within 135 min of visible light exposure was negligible in the absence of a catalyst, indicating that the direct photolysis of 2,4-D alone was insignificant. The removal of 2,4-D increased significantly in the presence of pristine HC and ZnBi-LDO. Within 135 min of visible light exposure, 64.5% and 59.3% of 2,4-D were removed in the presence of HC and ZnBi-LDO, respectively. The photocatalytic activity of the HC-ZnBi-LDO heterojunction increased significantly upon the introduction of a reasonable percentage of HC into ZnBi-LDO. The photocatalytic activity of the HC-ZnBi-LDO heterojunction increased further as the percentage of HC increased; however, the photocatalytic activity of the HC-ZnBi-LDO heterojunction decreased as the percentage of HC increased. The percentage of 2,4-D removed within 60 min of adsorption and 135 min of visible light exposure was 67.8% for 1% HC-ZnBi-LDO, 90.8% for 2% HC-ZnBi-LDO, and 52.9% for 5% HC-ZnBi-LDO. The change in the activity of the HC-ZnBi-LDO heterojunction under visible light may be explained as follows: when the amount of HC in HC-ZnBi-LDO was large enough, a sufficient number of photoinduced electron-hole pairs were generated, resulting in an increased efficiency of 2,4-D degradation; however, an excessive loading of HC caused HC to act as a bridge for the recombination of photoinduced electron-hole pairs or significantly decreased the efficiency at the junctions between HC and ZnBi-LDO in the heterojunction, thereby interfering with the charge transfer at these junctions. Among the photocatalysts, the 2% HC-ZnBi-LDO heterojunction exhibited superior photocatalytic activity. 

The results obtained in this study indicate that there was close contact between the surfaces of HC and ZnBi-LDO, which enabled the formation of the HC-ZnBi-LDO heterogeneous catalyst. The 2,4-D photodegradation data were matched to a pseudo-first-order kinetics equation, which was represented as follows:
ln(*C*_o_/*C_t_*) = *kt,*
where *t* is the degradation time (min), *k* is the apparent rate constant for 2,4-D degradation (min^−1^), and *C_o_* and *C_t_* are the concentrations of 2,4-D (mg/L) after dark adsorption equilibrium (*t* = 0) and at a certain time exposed to visible light (*t* = t), respectively.

Thus, the observed apparent rate constants (*k*) were calculated, as shown in Figure 2b. Notably, the apparent rate constant of 2,4-D degradation over the heterojunction increased considerably. The photocatalytic activity of the heterojunction 2% HC-ZnBi-LDO was highest, with *k* = 0.015 min^−1^, and the efficiency of degradation was approximately 2.6 and 10.7 times higher than those of pristine ZnBi-LDO (*k* = 0.0059 min^−1^) and HC (*k* = 0.0014 min^−1^), respectively. Because 2% HC-ZnBi-LDO had the best photocatalytic activity, it was used to study factors that could affect the efficiency of 2,4-D degradation within 135 min of visible light exposure.

To confirm that 2% HC-ZnBi-LDO had the best photocatalytic activity, photoluminescence (PL), UV-Vis DRS, and XPS analyses were performed.

As is well known, recombination of photoinduced electron-hole pairs normally emits a PL signal, while maximum separation of photoelectron–hole pairs suppresses the PL signal [28]. The PL spectra of ZnAl-LDO and 2%HC-ZnBi-LDO samples excited at 370 nm (Figure 3a) showed that 2%HC-ZnAl-LDO displayed similar but significantly lower luminescence properties than ZnBi-LDO, which may be due to a smaller extent of photoinduced electron-hole recombination. This phenomenon confirms that the improvement of the photocatalytic activity of 2%HC-ZnBi-LDO is due to the efficient suppression of photoinduced electron-hole recombination upon the formation of a heterojunction between HC and ZnBi-LDO.

Semiconductors have a characteristic band energy that absorbs a precise frequency of light, although the position of the band edge depends on the surface charge [29]. As shown in Figure 3b, the pristine ZnBi-LDO material exhibited a visible light response with the two absorption band edges at 425 and 480 nm, which were associated with ZnO and Bi_2_O_3_, respectively. The absorption edges of 2% HC-ZnBi-LDO were the same as those of pristine ZnBi-LDO and were redshifted relative to those of pristine ZnBi-LDO. Therefore, the strong association of HC with ZnBi-LDO caused the widening of the light absorption window toward the visible light region.

The calculated band gaps (Eg) of the pristine HC, ZnBi-LDO, and 2% HC-ZnBi-LDO samples are shown in Figure 3b (inset). The band gaps of ZnBi-LDO and 2% HC-ZnBi-LDO were derived from the plot of (αhυ)^2^ vs. energy (hυ) 3.05 eV and 2.95 eV, respectively. Since the band gap of 2% HC-ZnBi-LDO was smaller than that of ZnBi-LDO, the photocatalytic activity of 2% HC-ZnBi_LDO was significantly increased compared to that of ZnBi-LDO.

The XPS spectra of Zn, Bi, O, and C of ZnBi-LDO and 2% HC-ZnBi-LDO are shown in Figure 3c. The two peaks at binding energies of 1044.7 and 1021.6 eV were attributed to Zn 2p1/2 and Zn 2p3/2 in ZnBi-LDO, respectively, representing typical valence state peaks of Zn^2+^ [30]. However, a 0.6–0.7 eV shift to lower binding energies of Zn 2p for the 2% HC-ZnBi-LDO sample compared with that of pristine ZnBi-LDO was observed. This could be due to the strong interaction between metals in ZnBi-LDO with carboxyl groups (COO^−^) of HC forming Zn-O-C and Bi-O-C bonds, which promotes transfer of electrons between ZnBi-LDO and HC. Two strong peaks located at the binding energies of 158.4 and 163.7 eV were assigned to Bi 4f7/2 and Bi 4f5/2 in ZnBi-LDO, representing oxidation state +3 (Bi^3+^) [30]. However, the Bi 4f peaks in 2% HC-ZnBi-LDO moved to lower binding energies of 157.9 and 163.2 eV, indicating the interaction between ZnBi-LDO and HC in heterojunctions. For ZnBi-LDO, the O 1s spectrum at 530.5 eV was attributed to the OH species surrounded by Zn/Bi atoms on the surface of ZnBi-LDO [31]. Additionally, the O 1s peaks of 2% HC-ZnBi-LDO could be separated into two strong peaks at binding energies of 530 eV and 528.5 eV, which were attributed to O^1−^ ions and O^2−^ ions [32]. The C 1s peaks of 2% HC-ZnBi-LDO could be deconvoluted into strong peaks at binding energies of 283.9, 285.4, and 287.6 eV, which refer to three types of carbon bonds: sp^2^ C (C–C/C=C), C-OH, and C=O, respectively [33,34,35,36].

#### 3.2.2. Effect of pH

It is acknowledged that the pH of the solution affects photocatalytic degradation, and the optimal pH depends mainly on the type of pollutant to be treated [37]. The initial pH of the solution varied between pH 2 and 7. These experiments were conducted at an initial concentration of 30 mg/L of 2,4−D, with a 2% HC-ZnBi_2_O_4_ dose of 1.0 g/L. The results presented in Figure 4a indicate that at a pH of 4, >90% of the herbicide, 2,4-D, was removed after 60 min of adsorption in the dark and 135 min of visible light exposure. Any change to pH (pH > 4 or pH < 4) resulted in a lower removal of 2,4-D. It is acknowledged that the p*Ka* value of 2,4-D was 2.74, and therefore, 2,4-D existed in the molecular form at pH < 2.74 and in the anionic form at pH > 2.74. The maximum photodegradation efficiency of 2,4-D occurred at pH 4. This may be related to the p*Ka* of the herbicide, 2,4-D, and the pH_PZC_ of the photocatalyst. The pH can change the electric charge at the surface of the catalyst [37]. The point of zero charge (PZC) for 2% HC-ZnBi-LDO, measured in this study, was 6.65 (Figure 4b). The surface charge of 2% HC-ZnBi-LDO will be positive at a pH lower than pHpzc and negative at a pH higher than pH_PZC_. The efficiency of 2,4-D removal by 2% HC-ZnBi-LDO was low at pH 2 and 7 because, at pH 2, the contact between the 2,4-D molecule and the positive surface of the catalyst was limited; at pH 7, the anionic 2,4-D was repelled by the negative surface of the catalyst. Therefore, the solution pH must be lower than pHpzc and higher than the pKa of 2,4-D for the effective removal of 2,4-D by 2% HC-ZnBi-LDO because the electrostatic interaction between the 2,4-D anions and the positive surface charge of the catalyst increased. Upon considering the interaction of the catalyst surface with 2,4-D, pH 4 appeared to be the optimal pH for the effective removal of 2,4-D. Thus, all subsequent experiments were carried out at pH 4.

#### 3.2.3. Effect of 2% HC-ZnBi-LDO Dosage

The effect of the amount of catalyst on the removal of 2,4-D was investigated by varying the concentration of 2% HC-ZnBi-LDO between 0.5 and 2.0 g/L at pH 4 and an initial 2,4-D concentration of 30 mg/L. The results, as shown in Figure 4c, demonstrated that the photocatalytic efficiency of 2% HC-ZnBi-LDO increased as the amount of catalyst increased from 0.5 to 1.0 g/L. This may be due to an increase in the surface area and the number of available active sites in the photocatalyst for the removal of 2,4-D. The removal efficiency was reduced at concentrations > 1.0 g/L because the 2% HC-ZnBi-LDO particles agglomerated, thereby reducing the number of available active sites on the surface of 2% HC-ZnBi-LDO. Additionally, higher concentrations of 2% HC-ZnBi-LDO caused turbidity, which blocked the penetration of light [38,39,40], and ultimately reduced the removal efficiency.

#### 3.2.4. Effect of Initial 2,4-D Concentration

The effect of the initial concentration of 2,4-D on the removal efficiency was carried out by varying the 2,4-D concentration from 10 to 50 mg/L with a 2% HC-ZnBi-LDO catalyst loading of 1.0 g/L at pH 4, and the results are presented in Figure 4d. The percentage of 2,4-D removal under visible light decreased with an increase in the 2,4-D concentration. The maximum removal occurred for the 10 mg/L 2,4-D solution with an apparent rate constant of 0.0961 min^−1^. The minimum removal occurred for the 50 mg/L 2,4-D solution with an apparent rate constant of 0.0033 min^−1^. These trends may be related to the deposition of 2,4-D molecules on the surface of 2% HC-ZnBi-LDO, which saturates and thereby inactivates the surface over a period. Moreover, a constant amount of 2% HC-ZnBi-LDO was used in all the experiments; therefore, the number of active sites on 2% HC-ZnBi-LDO was constant. The ratio of 2% HC-ZnBi-LDO to the initial 2,4-D concentration decreased as the 2,4-D concentration increased; therefore, there was a decrease in the number of available active sites. Similar observations have been reported in the literature [41,42,43].

#### 3.2.5. Reusability of the 2% HC-ZnBi-LDO Photocatalyst

To clarify the practical applicability, the durability of the 2%HC-ZnBi-LDO photocatalyst was evaluated by a reuse experiment. The reuse experiment was conducted for four consecutive cycles. In this experiment, 1 g/L of 2% HC-ZnBi-LDO photocatalyst was added to a 30 mg/L 2,4-D solution at pH 4.0. After 60 min of adsorption and 135 min of visible light exposure, the 2% HC-ZnBi-LDO photocatalyst was centrifuged, cleaned with water and ethanol, and dried in an oven at 100 °C for 90 min, after which the catalyst was used for the next run. As shown in Table 1, the removal efficiency of 2,4-D decreased slightly from 90.8 ± 2.1% to 86.4 ± 2.5% after four cycles; this may be related to the inactivation of a fraction of the 2% HC-ZnBi-LDO photocatalyst. Thus, the recycled 2% HC-ZnBi-LDO displayed high efficiency for 2,4-D removal. 

From the reuse experiment, in parallel with the reusability assessment, the degree of 2,4-D mineralization by the 2%HC-ZnBi-LDO catalyst must also be evaluated. To clarify that the 2,4-D was mineralized instead of incompletely decomposed by the 2%HC-ZnBi-LDO catalyst, the organic carbon (TOC) in the reaction solution and on 2% HC-ZnBi-LDO catalyst before light-on and after light-off, as well as inorganic chloride in the reaction solution after light-off, were measured and presented in Table 1. After 4 cycles, >79.0–86.4% of TOC was removed from the 2,4-D solution, while approximately 86.4–90.8% of 2,4-D was degraded. In addition, more than 86.3–89.5% of organic chloride was converted to inorganic chloride (Table 1), which confirms the almost complete mineralization of 2,4-D after 60 min of adsorption and 135 min of visible light exposure at pH 4.0 using the 2% HC-ZnBi-LDO catalyst. The adsorption of 2,4-D and its degradation intermediates on the surface of the 2%HC-ZnBi-LDO catalyst was confirmed by FTIR analysis (Figure 1b-after). There was no residue on the catalyst surface, which confirmed complete mineralization. 

Table 2 lists a number of studies that have been performed recently on the removal of 2,4-D by a variety of photocatalysts. For comparison, the results obtained from the current study are also displayed. The results show that this work has relatively good performance compared to other catalysts, especially in the visible light range. 

#### 3.2.6. Effect of Scavengers on the Degradation and Proposed Degradation Mechanism

The oxidizing species responsible for the degradation of 2,4-D over the 2% HC-ZnBi-LDO photocatalyst exposed to visible light were conducted by injecting different scavengers into the reaction system. These scavengers are known to inhibit the formation of oxidizing species, which do not facilitate the degradation of 2,4-D. The experiments were carried out with the participation of scavengers, with an initial 2,4-D concentration of 30 mg/L at pH 4.0, and a 2% HC-ZnBi-LDO of 1.0 g/L. 

The effect of the different scavengers on the rate of degradation of 2,4-D is shown in Figure 5a. Approximately 65.2% of 2,4-D was removed using the 2% HC-ZnBi-LDO photocatalyst after 135 min of visible light exposure when *tert*-butanol (2.0 mM, a scavenger of OH^●^ radicals) was injected into the system (with *k* = 0.0076 min^−1^); this result indicates that OH^●^ radicals are not primary active species. On the contrary, 1.0 mM behaved as a photoinduced *h^+^* scavenger and reduced the efficiency of the removal of 2,4-D; the efficiency of the removal of 2,4-D decreased from 90.8% to 39.4% after 135 min of visible light exposure (*k* = 0.0026 min^−1^). The reduction in the removal of 2,4-D in the participation of p-benzoquinone (2 mM, a scavenger of O_2_^●−^ radicals) was greater than that in the participation of Na_2_EDTA. The photocatalytic activity of 2% HC-ZnBi-LDO decreased to 32% after 135 min visible light exposure (*k* = 0.0014 min^−1^) in the participation of the O_2_^●−^ scavenger (Figure 5a). Therefore, the extent of the involvement of the oxidizing species responsible for the degradation of 2,4-D over 2% HC-ZnBi-LDO followed the order: O_2_^●−^ radicals > photoinduced *h^−^* > OH^●^ radicals.

One of the central factors contributing to the significant enhancement of the photocatalytic activity of the heterojunctions between semiconductors with different band gaps is the maximum separation of the charge carriers and rapid charge transfer. The PL analyses indicate that 2% HC-ZnBi-LDO is an optimal heterojunction. The PL intensity of 2% HC-ZnBi-LDO was quenched due to the maximum separation of the charge carriers and a better charge transfer synergy between the heterojunction interfaces; these factors enhanced the performance of the photocatalyst. A possible mechanism involving potential charge separation, migration, and the degradation of 2,4-D upon exposure to visible light over 2% HC-ZnBi-LDO was suggested, as shown in Figure 5b. After exposure to visible light, ZnBi-LDO can be excited due to the energy band gap to generate photoinduced electron (*e^−^*)-hole (*h^+^*) pairs. The photoinduced holes can oxidize H_2_O to OH^●^ while the photoinduced electrons migrate quickly and efficiently to the HC. The generation of such reactive oxygen species (O_2_^●−^ radicals), from the reduction of the O_2_ molecules adsorbed on the surface of 2% HC-ZnBi-LDO by photogenerated electrons can be achieved during photocatalysis. The photoinduced holes, OH^●^ radicals, and O_2_^●−^ radicals can directly oxidize 2,4-D to CO_2_ and H_2_O (or other intermediates). The degradation of 2,4-D by 2% HC-ZnBi-LDO upon exposure to visible light is proposed to involve the generation of photoinduced *e^−^ - h^+^* pairs, leading to the formation of OH^●^ and O_2_^●−^ radicals (Equations (1)–(7)):2% HC-ZnBi-LDO + *hν* → 2% HC-ZnBi-LDO (*e^−^, h^+^)*(1)
2% HC-ZnBi-LDO *e^−^* + O_2_ → O_2_^●−^(2)
2% HC-ZnBi-LDO (*h^+^*) + 2H_2_O → OH^●^ + H^+^(3)
2% HC-ZnBi-LDO (*h^+^*) + 2,4-D → CO_2_ + H_2_O + Cl^−^ +...(4)
O_2_^●-^ + 2,4-D → CO_2_ + H_2_O + Cl^−^ +...(5)
OH^●^ + 2,4-D → CO_2_ + H_2_O + Cl^−^ +...(6)
*h^+^* + *e*^−^ → (*e*^−^, *h^+^*) (negligible recombination)(7)

## 4. Conclusions

In this work, a series of HC-ZnBi-LDO heterostructure photocatalysts was successfully synthesized via hydrothermal combination with co-precipitation and used for the degradation of 2,4-D upon exposure to visible light. The results indicated that the HC-ZnBi-LDO photocatalyst containing 2 wt% HC exhibited the maximum efficiency for the degradation of 2,4-D (30 mg/L) at pH 4.0 using 1.0 g/L of catalyst. After four consecutive cycles, the efficiency of the degradation of 2,4-D using the 2% HC-ZnBi-LDO photocatalyst was 86.4%, compared to an initial 90.8%. Approximately 79–86.4% of the TOC was removed in 135 min of visible light exposure during four consecutive cycles. The degree of the involvement of reactive species using the 2% HC-ZnBi-LDO photocatalyst upon exposure to visible light was O_2_^●−^ radicals > photoinduced *h^+^*> OH^●^ radicals. The incorporation of HC enhanced the photocatalytic activity of 2% HC-ZnBi-LDO by increasing both the visible light absorption and migration of photoinduced *e^−^*from ZnBi-LDO to HC, thus efficiently separating the photoinduced electron-hole pairs. The results of this study demonstrated that 2% HC-ZnBi-LDO can be used as a visible light photocatalyst for environmental remediation related to persistent organic pollutants.

## Figures and Tables

**Figure 1 materials-16-05027-f001:**
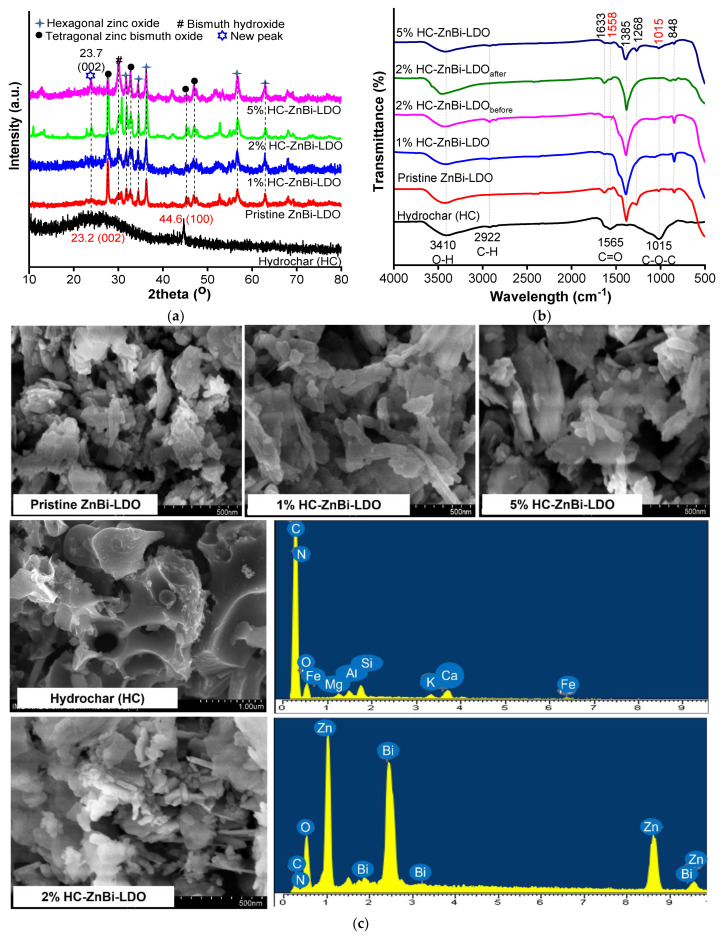
(**a**) XRD patterns; (**b**) FTIR spectra; and (**c**) Typical SEM image and EDS analysis of as-prepared samples.

**Figure 2 materials-16-05027-f002:**
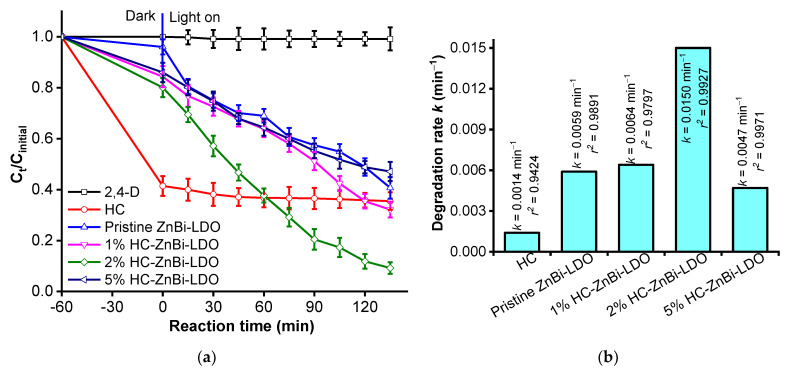
(**a**) The effects of different loading amounts of HC in ZnBi-LDO and (**b**) Rate constant *k* values of all the photocatalysts after 135 min of visible light exposure (catalyst: 1.0 g/L, 2,4-D: 30 mg/L at pH 4.0).

**Figure 3 materials-16-05027-f003:**
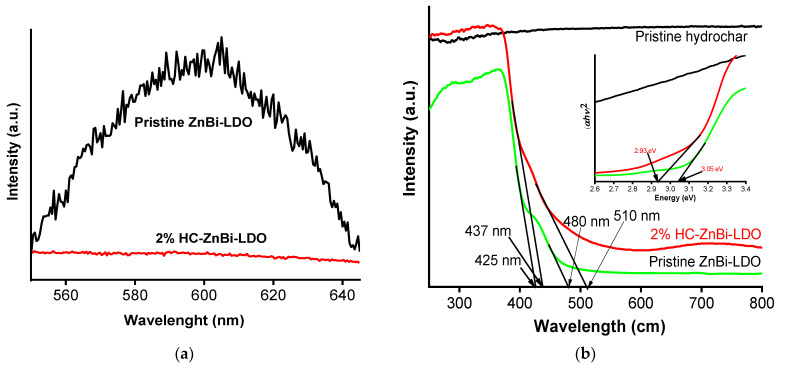
(**a**) PL emission spectra for the ZnBi-LDO and 2% HC-ZnBi-LDO; (**b**) UV−Vis diffuse reflectance spectra and band gap calculation using Tauc’s plot of as-prepared samples (inset) and (**c**) XPS spectra of the ZnBi-LDO and 2% HC-ZnBi-LDO (Zn2p, Bi4f, O1s, and C1s spectra).

**Figure 4 materials-16-05027-f004:**
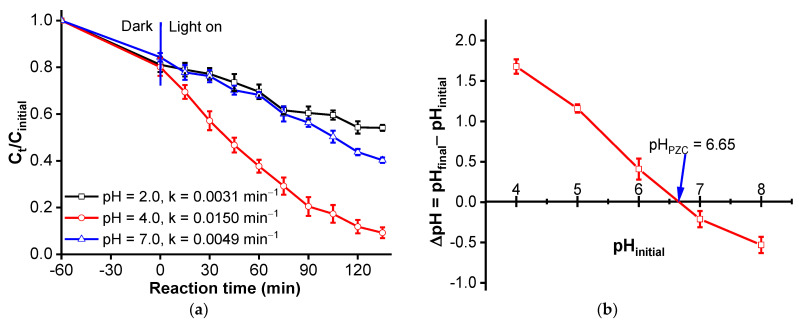
(**a**) The effects of pH solution on degradation of 2,4-D over 2% HC-ZnBi-LDO photocatalyst; (**b**) Point of zero charge (pH_PZC_) of 2% HC-ZnBi-LDO photocatalyst (ΔpH versus pH initial); (**c**) The effects of the amount of 2% HC-ZnBi-LDO photocatalyst on degradation of 2,4-D and (**d**) The effects of initial 2,4-D concentration on degradation of 2,4-D over 2% HC-ZnBi-LDO photocatalyst after 135 min of visible light exposure.

**Figure 5 materials-16-05027-f005:**
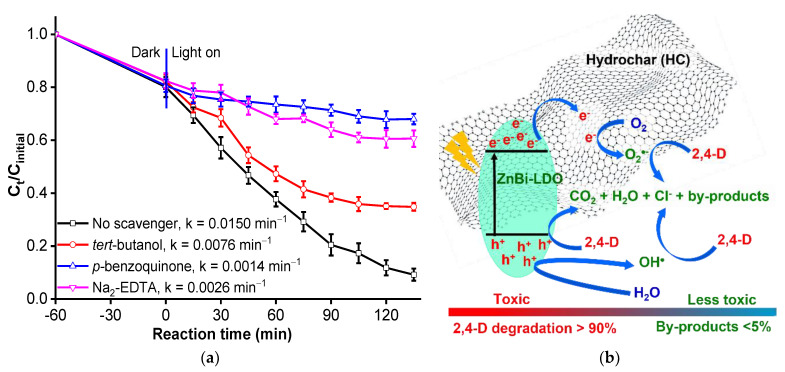
(**a**) The effects of various scavengers (*tert*-butanol, Na_2_EDTA and p-benzoquinone) on the photodegradation of 2,4-D over 2% HC-ZnBi-LDO after 135 min of visible light exposure and (**b**) Schematic diagram of the photoinduced *e^−^—h^+^* separation and transfer of photoinduced *e^−^*at the visible light-driven 2% HC-ZnBi-LDO interface.

**Table 1 materials-16-05027-t001:** Durability of 2% HC-ZnBi-LDO photocatalyst and degree of mineralization of 2,4-D over 2% HC-ZnBi-LDO photocatalyst (initial TOC concentration of 13.07 ± 0.23 corresponds to a solution of 2,4-D of 30 mg/L; quantity of catalyst of 1.0 g/L; the amount of chloride in the molecular formula 2,4-D is theoretically estimated to be approximately 9.67 mg/L; initial TOC in 2% HC-ZnBi-LDO is not detected).

Recycling	TOC in HC-ZnBi-LDO (mg/g)		TOC in Solution before Visible Light-On		TOC in Solution after Visible Light-Off		%TOC Removal	Dechlorination (Inorganic Cl^−^)		%Degradation
	Before Light-On	After Light-Off	mg/L	%	mg/L	%		mg/L	%	
1st	2.60 ± 0.14	ND	10.41 ± 0.22	79.7 ± 1.7	1.77 ± 0.15	13.6 ± 1.2	86.4 ± 1.7	8.65 ± 0.23	89.5 ± 2.3	90.8 ± 2.1
2nd	2.38 ± 0.19	ND	10.73 ± 0.29	82.1 ± 2.2	2.18 ± 0.13	16.7 ± 1.0	83.3 ± 1.0	8.56 ± 0.16	88.6 ± 1.7	89.8 ± 1.3
3rd	1.87 ± 0.18	ND	11.26 ± 0.23	86.1 ± 1.7	2.44 ± 0.18	18.7 ± 1.4	81.3 ± 1.4	8.50 ± 0.28	87.9 ± 2.9	88.2 ± 1.7
4th	1.72 ± 0.11	ND	11.42 ± 0.17	87.4 ± 1.3	2.75 ± 0.22	21.0 ± 1.7	79.0 ± 1.7	8.34 ± 0.19	86.3 ± 1.9	86.4 ± 2.5

ND: not detected, LOD = 4 ppb = 4 × 10^−6^ mg/g.

**Table 2 materials-16-05027-t002:** Comparison of degradation efficiency for different photocatalysts.

Catalyst	Light Source	2,4-D (mg/L)	Amount of Catalyst (g/L)	Time (min)	pH	Degradation (%)	Mineralization (%)	Ref.
Current study	Visible	30	1.0	135	4.0	90.8	86.4	
TiO_2_@MgFe_2_O_4_/H_2_O_2_	Visible	100	0.5	240	2.0	>83.0	-	[15]
rGO/ZnBi_2_O_4_	Visible	30	1.0	120	2.45	>90.0	83.7	[21]
Ag_3_PO_4_/TiO_2_	Visible	10	1.0	60	3.0	98.4	-	[44]
Ag/RGO-TiO_2_	Solar	10	-	160	-	100	-	[45]
Mn-ZnO/graphene	LED	25	2.0	120	5.0	66.2	-	[28]
TiO_2_/zeolite HY	UV	200	2.0	300	3.0	100.0	>80.0	[46]
TiO_2_	UVA	25	1.5	180	5.0	97.5	39.9	[32]
TiO_2_ + H_2_O_2_	UVA	25	1.5	180	5.0	99.7	56.0	[32]

## Data Availability

The data used to support the findings of this study are available from the corresponding author upon request.

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
