# Peer review of "Key Role of Corncob Based-Hydrochar (HC) in the Enhancement of Visible Light Photocatalytic Degradation of 2,4-Dichlorophenoxyacetic Acid Using a Derivative of ZnBi-Layered Double Hydroxides"

_materials, 2023, doi:10.3390/ma16145027_

Round 1

Reviewer 1 Report

In the present paper, the addition of hydrochar on derivative of ZnBi-layered double hydroxides was found to significantly enhance its photocatalytic degradation of 2,4-dichlorophenoxyacetic acid. The promoting effect was attributed to the rapid separation and migration of charge carriers at the interfaces between HC and ZnBi-LDO. The results are interesting and have important implications for application of hydrochar. This paper can be published after minor revision. Detailed comments are as follows.

1)      Since no clear peaks appear, it can’t be concluded that HC displays absorption over the visible range, which extends to the infrared region.

2)      It is better to list the comparison results between the present work and previous ones in one table.

3)      Full name should be given when it first appears, e.g. 2,4-dichlorophenoxyacetic acid.

Author Response

Ref.: materials-2504290

Key role of corncobs based-hydrochar (HC) in the enhancement of visible-light photocatalytic degradation of 2,4-D using derivative of ZnBi-layered double hydroxides

Revised name as Reviewer 1 suggested “full name should be given when it first appears”:

Key role of corncobs based-hydrochar (HC) in the enhancement of visible-light photocatalytic degradation of 2,4-dichlorophenoxyacetic acid using derivative of ZnBi-layered double hydroxides

In this document, the words with color in black are reviewer’s comments. The words with color in blue are authors’ reply, and the words with color in red is revised words in the text.

Reviewer 1

In the present paper, the addition of hydrochar on derivative of ZnBi-layered double hydroxides was found to significantly enhance its photocatalytic degradation of 2,4-dichlorophenoxyacetic acid. The promoting effect was attributed to the rapid separation and migration of charge carriers at the interfaces between HC and ZnBi-LDO. The results are interesting and have important implications for the application of hydrochar. This paper can be published after minor revision. Detailed comments are as follows.

Comment 1: Since no clear peaks appear, it can’t be concluded that HC displays absorption over the visible range, which extends to the infrared region.

Response: Thanks for the comment. The sentence “HC displays absorption over the visible range, which extends to the infrared region” has been removed.

Comment 2: It is better to list the comparison results between the present work and previous ones on one table.

Response: Thanks for the comment. The comparison results are listed in Table 2.

Table 2. Comparison of degradation efficiency for different photocatalysts

Catalyst

Light source

2,4-D (mg/L)

Amount of catalyst (g/L)

Time (min)

pH

Degradation (%)

Mineralization

(%)

Ref.

Current study

Visible

30

1.0

135

4.0

90.8

86.4

TiO2@MgFe2O4/H2O2

Visible

100

0.5

240

2.0

>83.0

-

[15]

 rGO/ZnBi2O4

Visible

30

1.0

120

2.45

>90.0

83.7

[21]

Ag3PO4/TiO2

Visible

10

1.0

60

3.0

98.4

-

[44]

Ag/RGO-TiO2

Solar

10

-

160

-

100

-

[45]

Mn-ZnO/graphene

LED

25

2.0

120

5.0

66.2

-

[28]

TiO2/zeolite HY

UV

200

2.0

300

3.0

100.0

>80.0

[46]

TiO2

UVA

25

1.5

180

5.0

97.5

39.9

[32]

TiO2+ H2O2

UVA

25

1.5

180

5.0

99.7

56.0

[32]

Comment 3: Full name should be given when it first appears, e.g. 2,4-dichlorophenoxyacetic acid.

Response: Thanks for the comment. The problems you mention have been fixed.

Reviewer 2 Report

To ameliorate the content of this article, I give some suggestions on :

1. Abstract : a sentence " Further, .....

2. Introduction :  as a new paragraph for : " In this study, ...."

3. Resulta and Discussion : in Page (4/15), add more information regarding on the FTIR analysis after photodegrading. in page 6/15 (XPS analysis): need more pronounced explanation regarding on the types of interaction between ZnBi-LDO and HC

All the suggestion is attached on the article.

Author Response

Ref.: materials-2504290

Key role of corncobs based-hydrochar (HC) in the enhancement of visible-light photocatalytic degradation of 2,4-D using derivative of ZnBi-layered double hydroxides

Revised name as Reviewer 1 suggested “full name should be given when it first appears”:

Key role of corncobs based-hydrochar (HC) in the enhancement of visible-light photocatalytic degradation of 2,4-dichlorophenoxyacetic acid using derivative of ZnBi-layered double hydroxides

In this document, the words with color in black are reviewer’s comments. The words with color in blue are authors’ reply, and the words with color in red is revised words in the text.

Reviewer 2

To ameliorate the content of this article, I give some suggestions on:

Comment 1: Abstract: a sentence " Further, .....

Response: Thanks for the comment, the sentence has been fixed as

“In addition, more than 86.4% of 30 mg/L 2,4-D was mineralized by 2% HC-ZnBi-LDO and the activity of 2% HC-ZnO-LDO was remarkably stable after 4 consecutive reuses”.

Comment 2:  Introduction:  as a new paragraph for: " In this study, ...."

Response: Thanks for the comment, the problem has been fixed.

Comment 3: Results and Discussion:

In Page (4/15), add more information regarding the FTIR analysis after photodegrading:

Response: Thanks for the comment, information regarding the FTIR analysis after photodegrading has been added in Figure 1b (2%HC-ZnBi-LDO after degradation).

The adsorption of 2,4-D and its degradation intermediates on the surface of the 2%HC-ZnBi-LDO catalyst was confirmed by FTIR analysis (Figure 1 b-after). There was no residue on the catalyst surface, which confirmed complete mineralization.

Figure 1b

In page 6/15 (XPS analysis): need more pronounced explanation regarding on the types of interaction between ZnBi-LDO and HC

Response: Thanks for the comment, the explanation has been more pronounced as you suggested:

This could be due to the strong interaction between metals in ZnBi-LDO with carboxyl groups (COO-) of HC forming Zn-O-C and Bi-O-C bonds,

Reviewer 3 Report

Manuscript ID: materials-2504290

Title: Key role of corncobs based-hydrochar (HC) in the enhancement of visible-light photocatalytic degradation of 2,4-D using derivative of ZnBi-layered double hydroxides

Dear Authors

This manuscript reports the evaluation of the fabricated HC-ZnBi-LDO heterojunction for degradation of 2,4-D under visible light irradiation. Further, the influences of various operating parameters, such as pH of solution, catalyst dosage, and initial 2,4-D concentration, were considered to achieve the optimal decomposition of 2,4-D. Although, interesting results have been obtained, overall, the manuscript needs Major revision before it could be accepted for publication in the Journal of Materials. In this regard, the author(s) should improve their work according to the following indications.

1.     Which intermediates can be formed during the degradation? draw the related mechanisms.

2.     The findings of the present work should be compared to results from other researchers by considering various parameters such as lamp intensity, degradation time, catalyst dosage used and the percentage efficiency attained for degrading 2,4-D.

3.     The reusability of the prepared samples is missing. please add such tests.

4.     It is better to add the sensitivity analysis of the experiments.

Sincerely yours

Minor editing of English language required.

Author Response

Ref.: materials-2504290

Key role of corncobs based-hydrochar (HC) in the enhancement of visible-light photocatalytic degradation of 2,4-D using derivative of ZnBi-layered double hydroxides

Revised name as Reviewer 1 suggested “full name should be given when it first appears”:

Key role of corncobs based-hydrochar (HC) in the enhancement of visible-light photocatalytic degradation of 2,4-dichlorophenoxyacetic acid using derivative of ZnBi-layered double hydroxides

In this document, the words with color in black are reviewer’s comments. The words with color in blue are authors’ reply, and the words with color in red is revised words in the text.

Reviewer 1

In the present paper, the addition of hydrochar on derivative of ZnBi-layered double hydroxides was found to significantly enhance its photocatalytic degradation of 2,4-dichlorophenoxyacetic acid. The promoting effect was attributed to the rapid separation and migration of charge carriers at the interfaces between HC and ZnBi-LDO. The results are interesting and have important implications for the application of hydrochar. This paper can be published after minor revision. Detailed comments are as follows.

Comment 1: Since no clear peaks appear, it can’t be concluded that HC displays absorption over the visible range, which extends to the infrared region.

Response: Thanks for the comment. The sentence “HC displays absorption over the visible range, which extends to the infrared region” has been removed.

Comment 2: It is better to list the comparison results between the present work and previous ones on one table.

Response: Thanks for the comment. The comparison results are listed in Table 2.

Table 2. Comparison of degradation efficiency for different photocatalysts

Catalyst

Light source

2,4-D (mg/L)

Amount of catalyst (g/L)

Time (min)

pH

Degradation (%)

Mineralization

(%)

Ref.

Current study

Visible

30

1.0

135

4.0

90.8

86.4

TiO2@MgFe2O4/H2O2

Visible

100

0.5

240

2.0

>83.0

-

[15]

 rGO/ZnBi2O4

Visible

30

1.0

120

2.45

>90.0

83.7

[21]

Ag3PO4/TiO2

Visible

10

1.0

60

3.0

98.4

-

[44]

Ag/RGO-TiO2

Solar

10

-

160

-

100

-

[45]

Mn-ZnO/graphene

LED

25

2.0

120

5.0

66.2

-

[28]

TiO2/zeolite HY

UV

200

2.0

300

3.0

100.0

>80.0

[46]

TiO2

UVA

25

1.5

180

5.0

97.5

39.9

[32]

TiO2+ H2O2

UVA

25

1.5

180

5.0

99.7

56.0

[32]

Comment 3: Full name should be given when it first appears, e.g. 2,4-dichlorophenoxyacetic acid.

Response: Thanks for the comment. The problems you mention have been fixed.

This manuscript reports the evaluation of the fabricated HC-ZnBi-LDO heterojunction for degradation of 2,4-D under visible light irradiation. Further, the influences of various operating parameters, such as pH of solution, catalyst dosage, and initial 2,4-D concentration, were considered to achieve the optimal decomposition of 2,4-D. Although, interesting results have been obtained, overall, the manuscript needs Major revision before it could be accepted for publication in the Journal of Materials. In this regard, the author(s) should improve their work according to the following indications.

Comment 1: Which intermediates can be formed during the degradation? draw the related mechanisms.

Response: Thanks for the comment. The goal of this study is to fabricate a photocatalyst that completely decomposes pollutants to CO2 and H2O. The results of total organic carbon (TOC) analysis showed that 2,4-D was almost completely decomposed. Furthermore, the FTIR analysis of 2% HC-ZnBi-LDO photocatalyst after degradation (Figure 1b-after) indicated that there was no residue on the 2% HC-ZnBi-LDO surface, which confirmed complete mineralization. Therefore, this study did not investigate intermediate products of the degradation process.

Figure 1b

Comment 2: The findings of the present work should be compared to results from other researchers by considering various parameters such as lamp intensity, degradation time, catalyst dosage used, and the percentage efficiency attained for degrading 2,4-D.

Response: Thanks for the comment. The comparison results are listed in Table 2.

Table 2. Comparison of degradation efficiency for different photocatalysts

Catalyst

Light source

2,4-D (mg/L)

Amount of catalyst (g/L)

Time (min)

pH

Degradation (%)

Mineralization

(%)

Ref.

Current study

Visible

30

1.0

135

4.0

90.8

86.4

TiO2@MgFe2O4/H2O2

Visible

100

0.5

240

2.0

>83.0

-

[15]

 rGO/ZnBi2O4

Visible

30

1.0

120

2.45

>90.0

83.7

[21]

Ag3PO4/TiO2

Visible

10

1.0

60

3.0

98.4

-

[44]

Ag/RGO-TiO2

Solar

10

-

160

-

100

-

[45]

Mn-ZnO/graphene

LED

25

2.0

120

5.0

66.2

-

[28]

TiO2/zeolite HY

UV

200

2.0

300

3.0

100.0

>80.0

[46]

TiO2

UVA

25

1.5

180

5.0

97.5

39.9

[32]

TiO2+ H2O2

UVA

25

1.5

180

5.0

99.7

56.0

[32]

Comment 3: The reusability of the prepared samples is missing please add such tests.

Response: Thanks for the comment. Reusability of the most efficient photocatalyst (2%HC-ZnBi-LDO) has been presented in section 3.2.5. Reusability of the 2% HC-ZnBi-LDO photocatalyst

Comment 4: It is better to add the sensitivity analysis of the experiments.

Response: Thanks for the comment. Because TOC on 2% HC-ZnBi-LDO photocatalyst was not detect ed (ND) after light off (table 1) therefore, sensitivity of TOC analysis in solid samples was added as

“LOD = 4 ppb = 4´10-6 mg/g”

The manuscript has also been carefully corrected in English usage by a native professional editor (http://www.tandfeditingservices.com)

Reviewer 4 Report

The article is appropriate, engaging, and reasonable, but it needs corrections:
1. Catalyst recovery and reuse should be checked.
2. Why is xenon lamp not used?
3. It is better to compare the results of this study with other studies.
4. The abstract and conclusion should be written more numerically
5. Why is LCMSS analysis not considered for this task?
6. The following sources should be included in the article:

* Efficient purification of aqueous solutions contaminated with sulfadiazine by coupling electro-Fenton/ultrasound process: optimization, DFT calculation, and innovative study of …

* Heterogeneous Fenton-like Photocatalytic Process towards the Eradication of Tetracycline under UV Irradiation: Mechanism Elucidation and Environmental Risk Analysis

Author Response

Ref.: materials-2504290

Key role of corncobs based-hydrochar (HC) in the enhancement of visible-light photocatalytic degradation of 2,4-D using derivative of ZnBi-layered double hydroxides

Revised name as Reviewer 1 suggested “full name should be given when it first appears”:

Key role of corncobs based-hydrochar (HC) in the enhancement of visible-light photocatalytic degradation of 2,4-dichlorophenoxyacetic acid using derivative of ZnBi-layered double hydroxides

In this document, the words with color in black are reviewer’s comments. The words with color in blue are authors’ reply, and the words with color in red is revised words in the text.

The article is appropriate, engaging, and reasonable, but it needs corrections:

Comment 1: Catalyst recovery and reuse should be checked.

Response: Thanks for the comment. Catalyst recovery and reuse has been presented in section 3.2.5. Reusability of the 2% HC-ZnBi-LDO photocatalyst

Comment 2: Why is xenon lamp not used?

Response: Thanks for the comment. The idea of the experiment is to decompose pollution under visible light. Xenon or halogen lamps also emit visible light.) Currently, halogen lamps are available in our laboratory.

Comment 3: It is better to compare the results of this study with other studies.

Response: Thanks for the comment. The comparison results are listed in Table 2.

Table 2. Comparison of degradation efficiency for different photocatalysts

Catalyst

Light source

2,4-D (mg/L)

Amount of catalyst (g/L)

Time (min)

pH

Degradation (%)

Mineralization

(%)

Ref.

Current study

Visible

30

1.0

135

4.0

90.8

86.4

TiO2@MgFe2O4/H2O2

Visible

100

0.5

240

2.0

>83.0

-

[15]

 rGO/ZnBi2O4

Visible

30

1.0

120

2.45

>90.0

83.7

[21]

Ag3PO4/TiO2

Visible

10

1.0

60

3.0

98.4

-

[44]

Ag/RGO-TiO2

Solar

10

-

160

-

100

-

[45]

Mn-ZnO/graphene

LED

25

2.0

120

5.0

66.2

-

[28]

TiO2/zeolite HY

UV

200

2.0

300

3.0

100.0

>80.0

[46]

TiO2

UVA

25

1.5

180

5.0

97.5

39.9

[32]

TiO2+ H2O2

UVA

25

1.5

180

5.0

99.7

56.0

[32]

Comment 4: The abstract and conclusion should be written more numerically.

Response: Thanks for the comment. The percentage of 2,4-D and TOC treated has been added in abstract and conclusion.

Comment 5: Why is LCMS analysis not considered for this task?

Response: Thanks for the comment. LCMS was not considered for analysis in this study because the goal of this study was to fabricate a photocatalyst that completely degrades pollutants to CO2 and H2O. Total organic carbon (TOC) analysis results show that 2,4-D is almost completely decomposed. Furthermore, the FTIR analysis of 2% HC-ZnBi-LDO photocatalyst after degradation (Figure 1b-after) indicated that there was no residue on the 2% HC-ZnBi-LDO surface, which confirmed complete mineralization.

Figure 1b

Comment 6: The following sources should be included in the article:

* Efficient purification of aqueous solutions contaminated with sulfadiazine by coupling electro-Fenton/ultrasound process: optimization, DFT calculation, and innovative study of …

* Heterogeneous Fenton-like Photocatalytic Process towards the Eradication of Tetracycline under UV Irradiation: Mechanism Elucidation and Environmental Risk Analysis

Response: Thanks for the comment, the articles you suggested have been added to manuscript as Ref 10, 11

Round 2

Reviewer 3 Report

Dear Authors

I confirm that the authors have considered the comments and performed, thus this manuscript is acceptable in its current form.

Best regards.

Reviewer 4 Report

Meanwhile, don't be tired, dear writers
The edited manuscript is in my opinion acceptable for acceptance into print. Good luck